# Evaluation of Frost Impact on Traditional Ceramic Building Materials Utilized in Facing Walls

**DOI:** 10.3390/ma15165653

**Published:** 2022-08-17

**Authors:** Anna Kaczmarek, Maria Wesołowska

**Affiliations:** Faculty of Civil and Environmental Engineering and Architecture, Bydgoszcz University of Science and Technology, 85-796 Bydgoszcz, Poland

**Keywords:** microstructure, freeze–thaw durability, absorption, traditional ceramic building materials

## Abstract

This paper takes into consideration the performance of traditional bricks as part of a building exterior wall finish. Exterior wall materials change their properties when exposed to external environment. This process is extended over time and its intensity is closely related to microstructure, moisture and freeze–thaw cycles. Two methods of freeze–thaw durability tests were used in this study: standard and defined by the authors. The authors’ method incorporated the actual conditions of masonry unit function in exterior wall finish, i.e., cyclical effects of precipitation water, changes in temperature and air humidity. The laboratory test study included 50 freeze–thaw cycles. Three characteristic ranges of pore dimensions were indicated in the analysis: below 0.1 µm, between 0.1 and 3.0 µm and above 3 µm. Based on the method of freeze–thaw durability testing, the areas of microstructure changes were determined. The obtained results were related to the absorption of ceramic building materials. The authors’ method confirms the usage of traditional ceramic building materials designed for use in protected walls against water penetration in unprotected exterior wall finish. The critical water saturation method of masonry units (standard) based on extreme environmental conditions generates significant changes in porosity distribution that do not reflect real, i.e., moderate, conditions. This method is appropriate for masonry units operating in severe conditions, i.e., F2. The aim of this study is to suggest a methodology for durability tests of traditional ceramic masonry units to cyclic freezing and thawing, which are only exposed to F1 (moderate) conditions during operation. Changes in the microstructure of the ceramic building materials were used as the primary evaluation criterion. In order to determine the effect of cyclic temperature changes, the freeze–thaw durability test was performed according to generally accepted standard procedures and in-house methodology. The purpose of the study is to point out the individual approach for the analysis of the material–environment system. At the same time, it should inspire researchers to innovative methods which use external conditions in a laboratory environment.

## 1. Introduction

There is a large number of buildings in Poland, the elevations of which are made of masonry units that do not meet modern frost requirements. At the time of their erection, the walls were protected against blowing rain, soil moisture, etc. By their very existence, the walls made of traditional ceramic building materials testify to their long-term and stable adaptation to the environment. Although they do not meet modern durability requirements according to microexposure classes (Table 1), their retention state is good. This is due to the applied elevation and moisture barrier solutions. These walls are made of ceramic units that, according to the recommendations of EN 771-1 [1], are intended to be used in solutions protected against water penetration. The solutions operate in conditions defined by the standard as moderate—F1—with exposure to moisture and freeze–thaw cycles. Therefore, they do not have the possibility of full water saturation, combined with frequent freeze–thaw cycles due to climatic conditions and the lack of adequate protection generating F2 severe conditions. Therefore, it is worth asking a question as to whether plastering is necessary after many years of exposure without the protection required by Eurocode 6 [2], or whether such situations can be treated as facing walls. This question is particularly relevant in the context of intensification of degradation processes caused, among other things, by rapid climate change (intense rain, short-term snow cover periods, multiple 0 °C passes, sun exposure). An important factor here is the flow of water film down the wall surface in rain, which in the case of intensive precipitation forms a uniform layer constituting the basic source of moisture.

Important factors that cause the degradation of facing walls are freeze–thaw cycles of water and crystallization of soluble mineral salts [3,4,5,6,7]. In the case of subject literature, freeze–thaw durability is analyzed in two aspects [4,8,9]. The first group of tests focuses on macroscopic effects including weight loss, surface damage and strength parameters. The second group refers to the microstructure study strictly relating it to the resistance of ceramic building materials to frost corrosion [3,10,11,12,13].

In this case, pore dimensions are mentioned as an important parameter that focuses on three basic ranges:-Large pores, larger than 3.0 μm, that feature a favorable effect on the freeze–thaw durability [8,9,14]. This is due to the fact that large pores are a kind of compensation chamber for the stresses during ice crystallization [15]. Moreover, the crystallization pressure is lower in larger pores [16].-Medium-sized pores of 3.0 to 0.1 μm, which are considered to be critical pores that determine the freeze–thaw durability of the bricks [14,17].-Small pores of less than 0.1 μm, where the water freezing point is well below 0 °C [18].

The range of critical pores was narrowed by Elert et al. [19] to the scope of 0.2 μm to 2.0 µm. Culturone et al. [20] also found that pores smaller than 2.0 µm in diameter are responsible for water absorption and retention increase. Consequently, they determine the freeze–thaw durability of ceramic building materials. According to Tang et al. [21], the destructive effect occurs in pores with diameters below 1 µm, which increase in size under the influence of cyclic freezing and defrosting processes and thus cause a shift in the porosity structure towards larger pores (1.0 to 5.0 μm). Koroth [3] and Kung [22] had similar observations. According to [23], the shape and distribution of pores significantly affect the durability of masonry units There is no unified approach to the freeze–thaw durability test in the mentioned studies. At least four testing methods are indicated [14]:-Critical degree of water saturation of complete masonry units;-Critical degree of water saturation of crushed masonry units;-Exposure of masonry units to actual climatic conditions;-Freezing and thawing with sprinkling irrigation of test panels in accordance with EN 772-22 [24].

The method for determining the freeze–thaw durability depends on the adopted methodology and the number of freezing and thawing cycles. Frost corrosion results not only in mechanical property changes, but also in the way the material behaves in contact with water. These changes directly affect the moisture condition of the products in the walls.

In most cases, facing walls made of traditional ceramic building materials operate in an environment defined as F1 (moderate). A review of the current state of knowledge does not identify an individual approach to the issue. Taking into account the standard requirements, masonry units made of traditional ceramic building materials do not pass the freeze–thaw durability test. The authors suggest distinguishing the methods of freeze–thaw durability testing for elements operated in severe conditions and in moderate conditions. The aim of this study is to suggest a methodology for the durability tests of traditional ceramic masonry units to cyclic freezing and thawing, which are only exposed to F1 (moderate) conditions during operation. Changes in the microstructure of the ceramic building materials were used as the primary evaluation criterion. In order to determine the effect of cyclic temperature changes, the freeze–thaw durability test was performed according to generally accepted standard procedures and in-house methodology.

## 2. Materials

The bricks analyzed as part of the study are dated from the 1980s. At that time in Poland, masonry units were sourced from local brickyards, which used Hoffmann ring furnaces fired with pulverized coal. The technology was based on the mechanical product molding, using a plastic method and their natural drying, followed by burning at a temperature of +980 °C for 216 h. Bricks were made from clay materials which originated from local deposits.

The tested products belong to the group of masonry units to be used in walls protected against water penetration [1], with dimensions 250 × 120 × 65 mm (Figure 1). Their quality parameters are summarized in Table 2.

The tests were carried out on masonry units manufactured 35 years ago, stored in laboratory conditions (temp. 20 ± 2 °C and air humidity 50 ± 5%) in 6 samples.

The tests were conducted on 3 sample groups:-Output samples (A);-Samples after the standard freeze–thaw durability test (B);-Samples after the freeze–thaw durability test with authors’ method (C).

The order of tests was adapted to the crashed stages of the samples. In order to ensure the representation of the study results, each masonry unit was divided into 4 sections. One section was taken from each masonry unit. This resulted in laboratory samples consisting of 6 parts taken from 6 masonry units (Figure 2). In the first stage, the absorption tests were carried out with output samples (A) as well as freezing and thawing cycles employed for the standard method (sample B) and the authors’ method (sample C).

The samples were then used to determine the microstructure of each group (A, B and C). For this purpose, fragments of up to 1 cm^3^ were taken from the facing surface of each sample. A laboratory sample was selected with the quartering method from the obtained material (Figure 2). Quartering consisted of coning the collected and thoroughly mixed material, then flattening and cross-dividing it into 4 parts. Two diagonal parts were removed and the remaining two parts were re-mixed and the selection process was repeated. This procedure was performed 3 times to obtain a laboratory sample volume of about 5 cm^3^ that corresponds to the volume of the penetrometer tank.

## 3. Test Methods

### 3.1. Absorption Test

The absorption test was carried out with a soaking method of 6 brick samples. The samples were dried to a fixed weight at +40 °C, then weighed to the nearest 0.1%. The samples were placed in a vessel with supports of non-corrosive material. Then, they were flooded with water at room temperature up to ½ of their height, after 3 h the water was replenished up to the level of ¾ of the sample height and after 3 h the water was replenished again until they were completely submerged (Figure 3).

The samples were kept in water until their fixed weight was established. Samples were taken out individually for weighting purposes (this protects them from drying out). External surfaces of the samples were wiped with a damp cloth.

Sample absorption n_m_ was calculated from the Formula (1), in %:(1)$$n_{m}=\frac{C_{m}-C_{s}}{C_{s}}\times 100\;\left[\%\right],$$
where:

*C_m_*—weight of a water-soaked sample, g;

*C_s_*—weight of dried sample, g.

### 3.2. Freezing and Thawing Cycles

Freezing and thawing cycles were performed with two methods:-Critical degree of water saturation of masonry units;-The authors’ method.

In the first case, the samples were placed in a water container and saturated to a fixed weight. Then, the brick surfaces were dried from excess water and placed in a freeze–thaw durability test chamber (Figure 4). Fifty freeze–thaw cycles were assumed in the temperature range of –18 to +18 °C.

In the next step, the samples were placed in a climate chamber with the following operation cycles:-The transition period of 0.5 h, including temperature adjustment from baseline to –18 °C;-In total, 3.5 h at –18 °C;-The transition period of 0.5 h, including temperature adjustment from –18 °C to +18 °C and humidity at 90%;-In total, 3.5 h at +18 °C.

After each complete cycle, the samples were soaked again in water at +20 ± 2 °C according to the initial absorption procedure as described in EN 772-11 [24]. All 50 cycles were conducted this way.

### 3.3. Microstructure Test Using Mercury Intrusion Porosimetry

The microstructure test was performed using a 9500 series AutoPore IV mercury intrusion porosimeter equipped with two ports: low and high pressure with a maximum value of 33,000 psi (228 MPa), which allows measurements in the range of meso- and macropores. Before the actual test, the calibration and “blank test” of the penetrometer used in this test were carried out to determine volume, compressibility and thermal effect. An equilibrium time of 30 s was determined based on control measurements. As a result of the prepared sample test, the following parameters of the structure in question were determined: total pore volume, sample volume and its skeletal density, the distribution of the pore volume as a function of the pore diameter as integral and differential relation. There were 18 samples.

The share of the pore volume was calculated based on the formula:(2)$$U_{frost}=\frac{\sum_{i=0.1µm}^{\;\;\;\;\;3.0µm}IV_{frost}}{TIV}\cdot P,$$
(3)$$U_{nondest}=\frac{\sum_{i>3.0µm}IV_{nondest}}{TIV}\cdot P,$$
(4)$$U_{small}=\frac{\sum_{i<0.1µm}^{\;\;\;\;\;}IV_{small}}{TIV}\cdot P,$$
where:

*IV_nondest_*—% share of meso- and macropores larger than 3.0 µm in diameter;

*IV_frost_*—% share of mesopores with diameters in the range of 0.1 to 3.0 µm;

*IV_small_*—% share of nanopores smaller than 0.1 µm in diameter;

*P*—total porosity.

## 4. Test Results and Discussion

The results of the microstructure test indicate that, compared to the initial material, the samples tested in both with the standard and the authors’ method obtained lower values of total porosity (Table 3).

Lower value of porosity was obtained in the case of the material after treatment with the authors’ method of freeze–thaw durability tests (–1.94%). In the case of the critical degree of water saturation of masonry units, the difference was –0.67%. The factor that essentially differentiates the features of the material covered in this study is the dominant pore diameter and porosity structure. The initial material exhibits a bimodal arrangement, typical for ceramic building materials, with dominant pores of 1.3 μm and 0.045 μm in diameter (Figure 5). In the analyzed cases, there is one dominant diameter, the value of which is much higher (for the samples after the authors’ freeze–thaw durability test of 5 nm, and after the method of critical degree of water saturation of masonry units of 4.5 μm (Figure 5), respectively. The obtained dominant diameters above 3 μm are within the range to be considered safe from the point of view of frost damage. The differential curve plots of the pore size distribution of the tested brick samples are presented in Figure 6.

The form of frost damage is closely related to the pore distribution (Figure 6). In the initial material, the total porosity was 33.35% with a share of 15.95 percentage points from 0.1 to 3.0 μm in diameter. The remaining 10.56 percentage points were occupied with pores above 3 μm in diameter, and 6.84% with pores below 0.1 μm in diameter. In the samples that faced the authors’ freeze–thaw durability test, the total porosity was 31.41% with the pore share in the range of 0.1–3.0 μm, increasing to 18.03 percentage points. The share of pores with diameters greater than 3.0 μm is maintained at the same level. There was a significant decrease in the share of pores below 0.1 μm at 2.81 percentage points. For the method of critical water saturation degree of masonry units, the total porosity was 32.68%, with a similar share of pores in the range of 0.1–3.0 μm (16.20 percentage points). There was a significant increase in the share of pores above 3.0 μm by 3 percentage points compared to the initial material. The share of pores below 0.1 μm decreased three times to 2.14 percentage points. In all analyzed cases, frost damage increased the share of porosity in the range of 0.1 to 3.0 μm. With cyclic freezing and thawing, it was found that pores below 0.1 μm were also responsible for the porosity structure changes. Frost influence damages them, resulting in increased pore size.

Porosity changes affect the moisture properties of the tested ceramic building materials. The absorption of the samples increases in relation to the initial material (Table 3) by about 1% regardless of the method of freezing and thawing cycles used.

## 5. Conclusions

This paper analyzes the influence of cyclic freezing and thawing on selected properties of masonry units intended to be used in walls protected against water penetration. Two test methods were adopted to determine the freeze–thaw durability of these ceramic masonry units. One of them is the commonly used standard method of the critical degree of water saturation of masonry units and the other is the proposed authors’ method. The choice of the freeze–thaw durability test method should depend on the environmental conditions defined by the classes of microexposure and the intensity of the masonry unit exposure to moisture and freeze–thaw cycles. The standard test method—critical degree of water saturation of whole masonry units—is appropriate for severe conditions where it is possible. The authors’ proposed method is appropriate for moderate conditions where only partial saturation of the masonry unit with water takes place. Facing walls without a protective layer of plaster in most cases are only exposed to moderate conditions (F1), for which the standard method of freeze–thaw durability test is inadequate. The authors suggest distinguishing the methods of freeze–thaw durability testing for masonry units used as face walls.

In the case of elements operating in severe conditions, it is suggested to use the standard method, while for elements operating in moderate conditions the authors’ method described in the article is recommended.

According to the authors, the adopted solution is similar to the functioning of masonry units in the exterior surface finish. The obtained results confirm the usage of unprotected ceramic building materials designed for use in protected walls against water penetration.

Further studies are planned to analyze SEM, the distribution of soluble mineral salts and changes in the mechanical properties of the superficial ceramic layer. As a result of environmental influence, the processes of migration and crystallization of soluble mineral salts are also initiated, resulting in subsequent changes in the microstructure.

## Figures and Tables

**Figure 1 materials-15-05653-f001:**
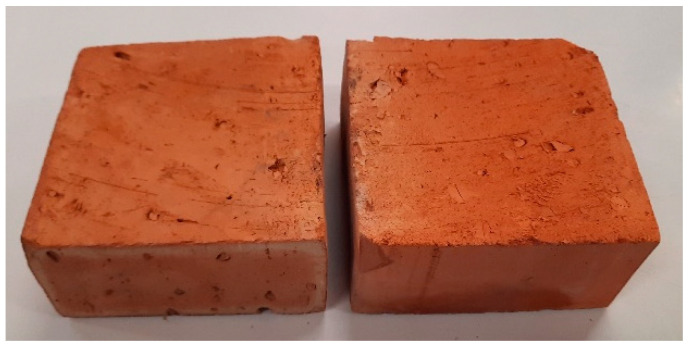
A brick sample used for tests.

**Figure 2 materials-15-05653-f002:**
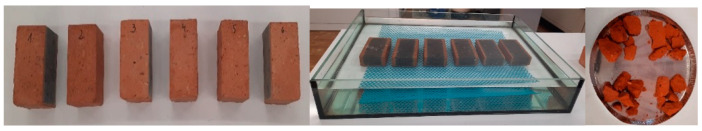
Preparation of samples for testing.

**Figure 3 materials-15-05653-f003:**
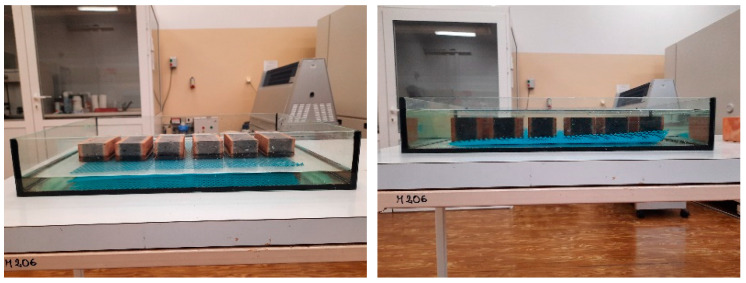
Absorption test.

**Figure 4 materials-15-05653-f004:**
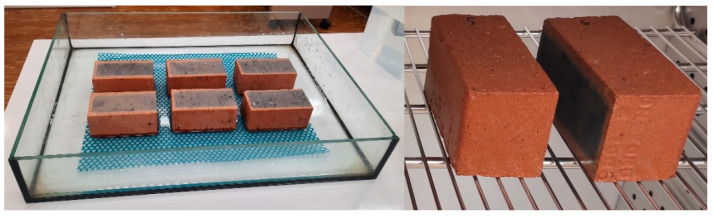
Sample conditioning in the authors’ freeze–thaw durability method.

**Figure 5 materials-15-05653-f005:**
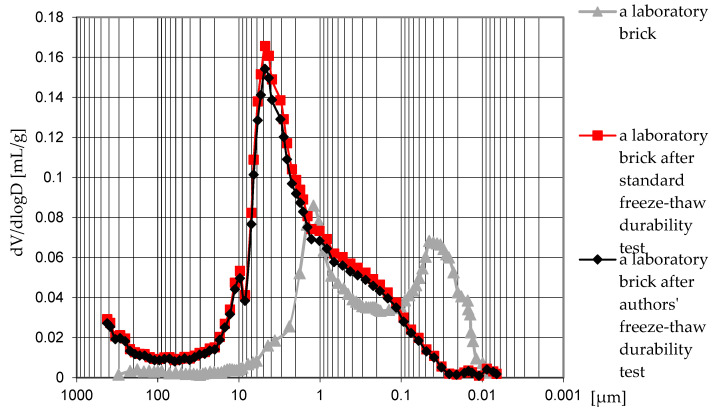
Differential curve of pore size distribution in test samples.

**Figure 6 materials-15-05653-f006:**
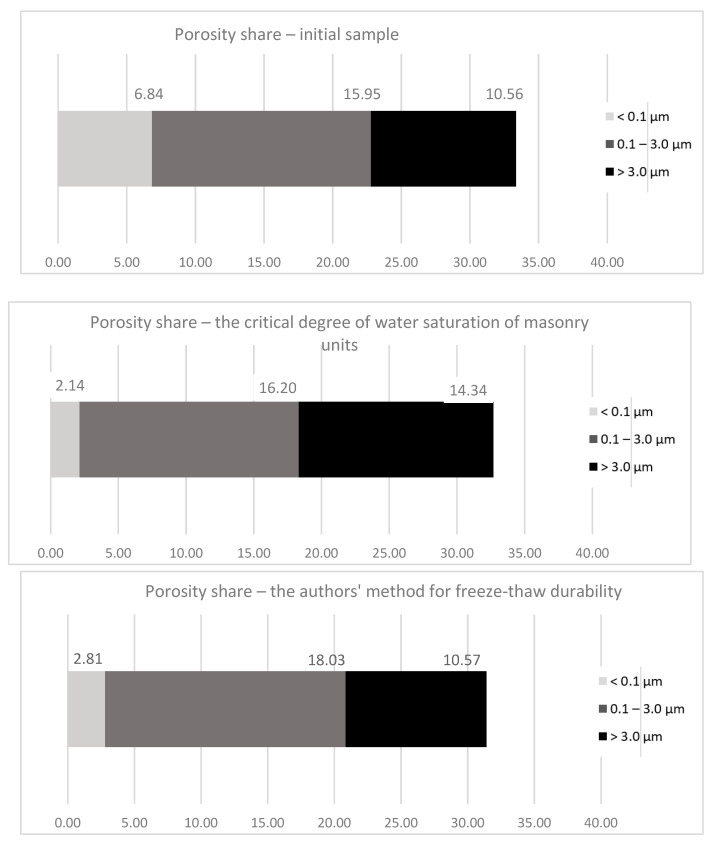
Typical porosity ranges in the tested clay bricks.

**Table 1 materials-15-05653-t001:** Classification of the microconditions of exposure of completed masonry [2].

Class	Micro Condition of the Masonry	Examples of Masonry in This Condition
MX 3.1	Exposed to moisture or wetting and freeze/thaw cycling but not exposed to external sources of significant levels of sulfates or aggressive chemicals.	Internal masonry exposed to high levels of water vapor, such as in a laundry. Masonry exterior walls sheltered by overhanging eaves or coping, not exposed to severe driving ram or frost. Masonry below frost zone in well drained non-aggressive soil. Exposed to freeze/thaw cycling.
MX 3.2	Exposed to severe wetting and freeze/thaw cycling but not exposed to external sources of significant levels of sulfates or aggressive chemicals.	Masonry not exposed to frost or aggressive chemicals. Location: in exterior walls with cappings or flush eaves; in parapets; in freestanding walls; in the ground; and under water. Exposed to freeze/thaw cycling.
MX 4	Exposed to saturated salt air, seawater or deicing salts.	Masonry in a coastal area. Masonry adjacent to roads that are salted during the winter.
MX 5	In an aggressive chemical environment.	Masonry in contact with natural soils or filled ground or groundwater, where moisture and significant levels of sulfates are present. Masonry in contact with highly acidic soils, contaminated ground or groundwater. Masonry near industrial areas where aggressive chemicals are airborne.

**Table 2 materials-15-05653-t002:** Quality parameters of ceramic building materials produced from the local Pliocene clay deposit.

Quality Parameters	Range	Average Value
batched water	(17.2–41.8)%	29.67%
drying contraction	(4.0–13.0)%	9.2%
absorption (temp. 980 °C)	(7.4–10.3)%	8.8%
compressive strength (temp. 980 °C)	(16.0–24.0) MPa	19.5 MPa
content of granular marl with more than 0.5 mm fraction	(0.00–0.40)%	0.031%
efflorescence of sulfate salts	-	none or minimal irremovable bloom

**Table 3 materials-15-05653-t003:** Tests results.

Test	Freezing and Thawing Cycles According to the Adopted Method		
	Output Sample	Critical Degree of Water Saturation of Masonry Units	Authors’ Method
general porosity [%]	33.35	32.68	31.41
absorption [%]	13.84	14.55	14.34
dimension of dominant pores [μm]	1.3, 0.045	4.5	5.0

## Data Availability

Data are contained within the article.

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
