# Peer review of "Evaluation of Frost Impact on Traditional Ceramic Building Materials Utilized in Facing Walls"

_materials, 2022, doi:10.3390/ma15165653_

Round 1

Reviewer 1 Report

The manuscript presents a new approach for freeze-thaw durability test on the masonry ceramic materials used in facing walls, and investigates the influence of cyclic freezing and thawing on the microstructure of ceramic materials. Some vital issues regarding the experimental procedures are needed to be carefully clarified before it can be considered for publication. The specific comments are as follows:

 1. Because the change in the pore distribution was selected to evaluate the freeze-thaw durability of masonry materials in this manuscript, the measures to ensure that the initial pore distributions of tested samples were approximately the same should be addressed in detail.

2. Figure 5, what did the symbol of white circle represent? Besides, the authors are recommended to explain why the difference in the test results obtained by the standard and the newly proposed methods was negligible.

3. Line 209, the Table 3 was not provided in the manuscript. Please check it.

4. Lines 216-217, the authors stated that the extreme environmental conditions considered in the standard method generated large change in pore distribution. However, the method proposed by authors also resulted in significant change in pore distribution according to the Fig. 6. In referee’s view, this conclusion was unreasonable.

5. Line 219, the authors concluded that the obtained results don’t reflect actual conditions. A clear description about the actual conditions was necessary for a better understanding of this conclusion.

6. If possible, the microstructure test, such as the scanning electron microscope (SEM) or the nuclear magnetic resonance (NMR), was recommended to investigate the transition mechanism of the pores with different sizes under the cyclic freezing and thawing.

Author Response

Response to Reviewer

  1. Because the change in the pore distribution was selected to evaluate the freeze-thaw durability of masonry materials in this manuscript, the measures to ensure that the initial pore distributions of tested samples were approximately the same should be addressed in detail.

The tests were carried out on masonry units manufactured 35 years ago, stored in laboratory conditions (temp. 20 ±2°C and air humidity 50 ±5%) in 6 samples.

The tests were conducted on 3 sample groups:

  • Output samples (A)
  • Samples after standard freeze-thaw durability test (B)
  • Samples after freeze-thaw durability test with authors' method (C)

The order of tests was adapted to the crashed stages of the samples. In order to ensure the representation of the study results, each masonry unit was divided into 4 sections. One section was taken from each masonry unit. This resulted in laboratory samples consisting of 6 parts taken from 6 masonry units (Fig. 2). In the first stage, the absorption tests were carried out with output samples (A) as well as freezing and thawing cycles employed for standard method (sample B) and authors' method (sample C).

The samples were then used to determine the microstructure of each group (A, B, and C). For this purpose, fragments of up to 1 cm3 were taken from the facing surface of each sample. A laboratory sample was selected with the quartering method from the obtained material (Fig. 2).

  1. Figure 5, what did the symbol of white circle represent? Besides, the authors are recommended to explain why the difference in the test results obtained by the standard and the newly proposed methods was negligible.

Thank you for your valuable insight. The figure has been corrected..

The authors agree that they compared the difference in the total sample porosity of the 1st size of dominant pores after freeze-thaw durability test with the standard method and the samples after freeze-thaw durability test with the authors' method. However, the share and distribution of pore size varied based on the adopted method of freeze-thaw durability test.  Therefore the authors suggest to distinguish methods of freeze-thaw durability testing for elements operated in severe conditions and in moderate conditions. In case of elements operating in severe conditions it is suggested to use the standard method, while for elements operating in moderate conditions the authors' method described in the article is recommended.

  1. Line 209, the Table 3 was not provided in the manuscript. Please check it.

Thank you for your valuable insight. Table 3 is referred to in the text.

  1. Lines 216-217, the authors stated that the extreme environmental conditions considered in the standard method generated large change in pore distribution. However, the method proposed by authors also resulted in significant change in pore distribution according to the Fig. 6. In referee’s view, this conclusion was unreasonable.

The authors thank you for your valuable insight. The conclusions have been amended. They are currently presented as follows:

  1. Line 219, the authors concluded that the obtained results don’t reflect actual conditions. A clear description about the actual conditions was necessary for a better understanding of this conclusion.

The choice of the freeze-thaw durability test method should depend on the environmental conditions defined by the classes of micro-exposure and the intensity of masonry unit exposure to moisture and freeze-thaw cycles. The standard test method – critical degree of water saturation of whole masonry units is appropriate for severe conditions where it is possible. The authors' proposed method is appropriate for moderate conditions where only partial saturation of masonry unit with water takes place. Facing walls without a protective layer of plaster in most cases are exposed only to moderate conditions (F1), for which the standard method of freeze-thaw durability test is inadequate.

  1. If possible, the microstructure test, such as the scanning electron microscope (SEM) or the nuclear magnetic resonance (NMR), was recommended to investigate the transition mechanism of the pores with different sizes under the cyclic freezing and thawing.

The authors thank you for your suggestions to expand the study. The proposed methods will be included in further studies.

Reviewer 2 Report

In this manuscript, the topic is interesting, but the following points must be carefully addressed before the paper is accepted.

The manuscript contains too many grammatical errors and hanging sentences. The manuscript needs to undergo professional language editing prior to submission

In the abstract, it should contain the aim/objective, methodology/experiments, findings, etc. Please rewrite the abstract.

The introduction has no proper of focus, the novelty and knowledge gap of study are not known and the objective of the study is not stated.

In the last paragraph of Introduction part, authors should justify their investigation. What can be done at the end of this study and what the results of this study provide for the literature?

Figure 3 have low resolution.

The results and the discussion part are not sufficient to study these samples. The discussion should be increased in this part.

Conclusions is not just about summarizing the key results of the study, it should highlight the insights and the applicability of your findings/results for further work, where is the recommendations for scientific community and possible using fields?

Author Response

Response to Reviewer

The manuscript contains too many grammatical errors and hanging sentences. The manuscript needs to undergo professional language editing prior to submission

In the abstract, it should contain the aim/objective, methodology/experiments, findings, etc. Please rewrite the abstract.

The abstract has been supplemented with the aim of the study.

The aim of this study is to suggest a methodology for durability tests of traditional ceramic masonry units to cyclic freezing and thawing, which are only exposed to F1 (moderate) conditions during operation. Changes in the microstructure of the ceramic building materials were used as the primary evaluation criterion.  In order to determine the effect of cyclic temperature changes, freeze-thaw durability test was performed according to generally accepted standard procedures and in-house methodology. 

The introduction has no proper of focus, the novelty and knowledge gap of study are not known and the objective of the study is not stated.

In the last paragraph of Introduction part, authors should justify their investigation. What can be done at the end of this study and what the results of this study provide for the literature?

Based on the suggestion, the following has been supplemented:

In most cases, facing walls made of traditional ceramic building materials operate in an environment defined as F1 (moderate). A review of the current state of knowledge does not identify an individual approach to the issue. Taking into account the standard requirements, masonry units made of traditional ceramic building materials do not pass the freeze-thaw durability test. The authors suggest to distinguish methods of freeze-thaw durability testing for elements operated in severe conditions and in moderate conditions.

Figure 3 have low resolution.

The results and the discussion part are not sufficient to study these samples. The discussion should be increased in this part.

Conclusions is not just about summarizing the key results of the study, it should highlight the insights and the applicability of your findings/results for further work, where is the recommendations for scientific community and possible using fields?

The choice of the freeze-thaw durability test method should depend on the environmental conditions defined by the classes of micro-exposure (Tab. 1) and the intensity of masonry unit exposure to moisture and freeze-thaw cycles [1]. The standard test method – critical degree of water saturation of whole masonry units is appropriate for severe conditions where it is possible. The authors' proposed method is appropriate for moderate conditions where only partial saturation of masonry unit with water takes place. Facing walls without a protective layer of plaster in most cases are exposed only to moderate conditions (F1), for which the standard method of freeze-thaw durability test is inadequate. The authors suggest to distinguish methods of freeze-thaw durability testing for masonry units used as face walls.

In case of elements operating in severe conditions it is suggested to use the standard method, while for elements operating in moderate conditions the authors' method described in the article is recommended.

Reviewer 3 Report

The authors present two methods of freeze thaw durability test

standard's  method and authors' method ,

my observations are on the manuscript

The precision is needed for each method (does the standard method is better for medium pores or the authors' method is better for high pores diameter?)

- Discussion section is needed

Author Response

The precision is needed for each method (does the standard method is better for medium pores or the authors' method is better for high pores diameter?)

The choice of the freeze-thaw durability test method should depend on the environmental conditions defined by the classes of micro-exposure (Tab. 1) and the intensity of masonry unit exposure to moisture and freeze-thaw cycles [1]. The standard test method – critical degree of water saturation of whole masonry units is appropriate for severe conditions where it is possible. The authors' proposed method is appropriate for moderate conditions where only partial saturation of masonry unit with water takes place. Facing walls without a protective layer of plaster in most cases are exposed only to moderate conditions (F1), for which the standard method of freeze-thaw durability test is inadequate. The authors suggest to distinguish methods of freeze-thaw durability testing for masonry units used as face walls.

In case of elements operating in severe conditions it is suggested to use the standard method, while for elements operating in moderate conditions the authors' method described in the article is recommended.

Reviewer 4 Report

1. Page 3 line 55. I am not sure that Table 1 is necessary. All information may be found with the reference.

2. Page 3 line 59. «...large pores, larger than 3.0 μm...)». It is necessary to correct the text in Abstract (lines 14 and 15) because there are nm in Abstract for pore size, not the μm. I hope the authors may do a wright choice.

3. Pages 3,4 line 70. «According to Tang et al. [22], the destructive effect occurs in pores of diameters below 1 nm,». Please, check the pores size ones more, there should be μm, not the nm.

4. As for the reference [22], line 221, the name «...Acient Chinese bricks under environmental vicissitudes» should be changed “Pore structure of Ancient Chinese bricks under environmental vicissitudes”.

Author Response

Dear Reviewer,

Thank you for your valuable tips on the article.

All the suggestions contained in the review have been included in the text.

Round 2

Reviewer 1 Report

The authors have addressed most of my initial concerns. There are a couple of issues that remain:

 1. The response to comment #1 is unsatisfactory. In my opinion, the brick masonry unit is nonhomogeneous due to its component and manufacturing process. Even though the samples for the output test, the standard test and the author’s method come from the fragments of a same brick, it doesn’t ensure that the porosity of each tested samples is approximately the same. I still don’t see any measures to avoid the initial difference of samples in the revised manuscript.

2. I don’t find the amended conclusions for the comment #4 in the revised manuscript or in the cover letter.

3. Line 245, there are some typos.

Author Response

Response to Reviewer

  1. The response to comment #1 is unsatisfactory. In my opinion, the brick masonry unit is nonhomogeneous due to its component and manufacturing process. Even though the samples for the output test, the standard test and the author’s method come from the fragments of a same brick, it doesn’t ensure that the porosity of each tested samples is approximately the same. I still don’t see any measures to avoid the initial difference of samples in the revised manuscript.

comment #1

Because the change in the pore distribution was selected to evaluate the freeze-thaw durability of masonry materials in this manuscript, the measures to ensure that the initial pore distributions of tested samples were approximately the same should be addressed in detail.

We agree with the reviewer that ceramic materials, due to the composition and production process, are heterogenic. Therefore, we felt that the quartering method was appropriate for selecting a test sample. We adapted this method from the preparation of an aggregate sample for testing.

Quartering consisted of coning the collected and thoroughly mixed material, then flattening and cross-dividing it into 4 parts. Two diagonal parts were removed and the remaining two parts were re-mixed and the selection process was repeated. This procedure was performed 3 times to obtain laboratory sample volume of about 5 cm3 that corresponds to the volume of the penetrometer tank.

If, according to the reviewer's opinion, there is a more reliable way to prepare samples, we would be very happy to review it in order to improve the quality of our further studies.

  1. I don’t find the amended conclusions for the comment #4 in the revised manuscript or in the cover letter.

comment #4

Lines 216-217, the authors stated that the extreme environmental conditions considered in the standard method generated large change in pore distribution. However, the method proposed by authors also resulted in significant change in pore distribution according to the Fig. 6. In referee’s view, this conclusion was unreasonable.

The authors thank you for your valuable insight. The conclusions have been amended. They are currently presented as follows:

The choice of the freeze-thaw durability test method should depend on the environmental conditions defined by the classes of micro-exposure (Tab. 1) and the intensity of masonry unit exposure to moisture and freeze-thaw cycles [1]. The standard test method – critical degree of water saturation of whole masonry units is appropriate for severe conditions where it is possible. The authors' proposed method is appropriate for moderate conditions where only partial saturation of masonry unit with water takes place. Facing walls without a protective layer of plaster in most cases are exposed only to moderate conditions (F1), for which the standard method of freeze-thaw durability test is inadequate. The authors suggest to distinguish methods of freeze-thaw durability testing for masonry units used as face walls.

In case of elements operating in severe conditions it is suggested to use the standard method, while for elements operating in moderate conditions the authors' method described in the article is recommended.

  1. Line 245, there are some typos.

We'd like to thank you for pointing that out. The indicated typos have been corrected:

One of them is the commonly used standard method of critical degree of water saturation of masonry units, the other – the proposed authors’ method.

Reviewer 2 Report

The manuscript can be accepted for publication.

Author Response

Thank You for revision

Reviewer 3 Report

The authors gave the response awaiting 

Author Response

Thank You for revision.